# A single dietary factor, daily consumption of a fermented beverage, can modulate the gut bacteria and fecal metabolites within the same ethnic community

Santanu Das,[1,2] Ezgi Özkurt,[3,4] Tulsi Kumari Joishy,[1] Ashis K. Mukherjee,[1] Falk Hildebrand,[3,4] Mojibur R. Khan[1]

**ABSTRACT**  In this study, the impact of traditional rice-based fermented alcoholic beverages (two types of Apong drink: Poro and Nogin) on the gut microbiome and health of the Mishing community in India was examined. Two groups ($n = 71$ in each group, 58 females and 84 males) that consumed one of these beverages were compared to a control group ($n = 24$, all males) that did not consume either beverage. Gut microbial composition was analyzed by sequencing 16S rRNA of fecal metagenomes and analyzing untargeted fecal metabolites, and short-chain fatty acids (SCFAs). We also collected data on anthropometric measures and serum biochemical markers. Our results showed that Apong drinkers had higher blood pressure, but lower blood glucose and total protein levels than other non-drinkers. Also, gut microbiome composition was found to be affected by the choice of Apong, with Apong drinkers having a more diverse and distinct microbiome compared to non-drinkers. Apong drink type or being a drinker or not explained even a higher variation of fecal metabolome composition than microbiome composition and Apong drinkers had lower levels of the SCFA isovaleric acid than non-drinkers. Overall, this study shows that a single dietary factor can significantly impact the gut microbiome of a community and highlights the potential role of traditional fermented beverages in modulating gut bacteria.

**IMPORTANCE**  Our study investigated how a traditional drink called Apong, made from fermented rice, affects the gut and health of the Mishing community in India. We compared two groups of people who drink Apong to a group of people who do not drink it. To accomplish this, we studied the gut bacteria, fecal metabolites, and blood samples of the participants. It was found that the people who drank Apong had higher blood pressure but lower blood sugar and protein levels than people who did not drink it. We also found that the gut microbiome composition of people who drank Apong was different from those who did not drink it. Moreover, people who drank Apong had lower levels of isovaleric acid in their feces. Overall, this study shows that a traditional drink like Apong can affect the gut bacteria of a community.

**KEYWORDS**  gut microbiome, alcoholic beverage, fermented beverage, short-chain fatty acids, fecal metabolites

The gut microbiome is a collection of microorganisms in the human intestine that perform many important functions, including digestion, transformation of nutrients, and immune system regulation (1). A well-balanced gut microbiome composition can provide health benefits, while imbalances can lead to disorders related to metabolism and immunity (2). The gut microbiome is influenced by various factors, including diet, medication, and age. Diet is a particularly important factor that affects the gut microbiota composition and its interactions with the host (3–7). Fermented foods and

Address correspondence to Falk Hildebrand, falk.hildebrand@quadram.ac.uk, or Mojibur R. Khan, mojibur.khan@gmail.com.

Santanu Das and Ezgi Özkurt contributed equally to this article. Author order was determined on the basis of conceptualization of the study and CRediT roles.

The authors declare no conflict of interest.

See the funding table on p. 15.

beverages, such as yogurt, kefir, fermented cottage cheese, kimchi, and kombucha tea, are rich in microorganisms that can have an effect on the gut microbiome and increase overall microbial diversity (8–14).

Rice-based fermented beverages are an important part of the diet and cultural heritage of Mongoloid communities in South-East Asia (15). In Assam, India, the Mishing community consumes two types of rice-based alcoholic beverages called Poro Apong and Nogin Apong. Within the Mishing community, some subgroups consume only one type of Apong, despite having similar lifestyles and dietary habits. Poro Apong is prepared with roasted rice, ash of rice husk, and a dry starter cake which is usually rich in beneficial microorganisms (16, 17). The mixture is then allowed to undergo solid-state fermentation for 7–10 days. It is filtered through ash to produce a dark, clear liquid with physical and sensory properties similar to stout beer. Nogin Apong is made with steamed rice and has physical and sensory properties similar to Makgeolli, a fermented beverage from South Korea. We previously reported that the alcohol content of Apong is 9–11% (vol/vol) and is rich in phenolic contents (18). Nogin Apong had a diverse array of lactic acid bacteria and was rich in saccharides and amino acids, while Poro Apong was dominated by *Lactobacillus* (18). In this study, we aimed to determine how these two types of Apong may affect the gut microbiome in volunteers from the same ethnicity who have the same dietary habits but differ in their choice of Apong (Fig. 1).

## RESULTS

### Apong drinkers have distinct levels of biochemical markers compared to non-drinkers, including differences in blood pressure, glucose, protein levels, and liver enzymes

Most individuals in this study had normal physiological and biochemical test results regardless of their lifestyles and dietary habits. However, blood pressure was significantly higher in Poro drinkers compared to the controls. (Fig. 2A, a and b). Both Nogin and Poro drinkers had lower total protein and albumin levels in their blood than non-drinkers (Fig. 2A, c and d), with Nogin drinkers having lower albumin to globulin ratio (Fig. 2A, f). Both Nogin and Poro drinkers also had lower blood glucose levels than non-drinkers (Fig. 2A, g), which is consistent with the blood glucose-lowering properties of fermented foods and beverages (19). Lipid levels, as measured by triglycerides, were within the normal range and comparable in all participants (Fig. 2A, h). Poro drinkers had significantly lower levels of serum glutamic-pyruvic transaminase (SGPT; Fig. 2A, i) and Serum glutamic-oxa-loacetic transaminase (SGOT; Fig. 2A, j), which is a sign of a healthy liver (20), and lower

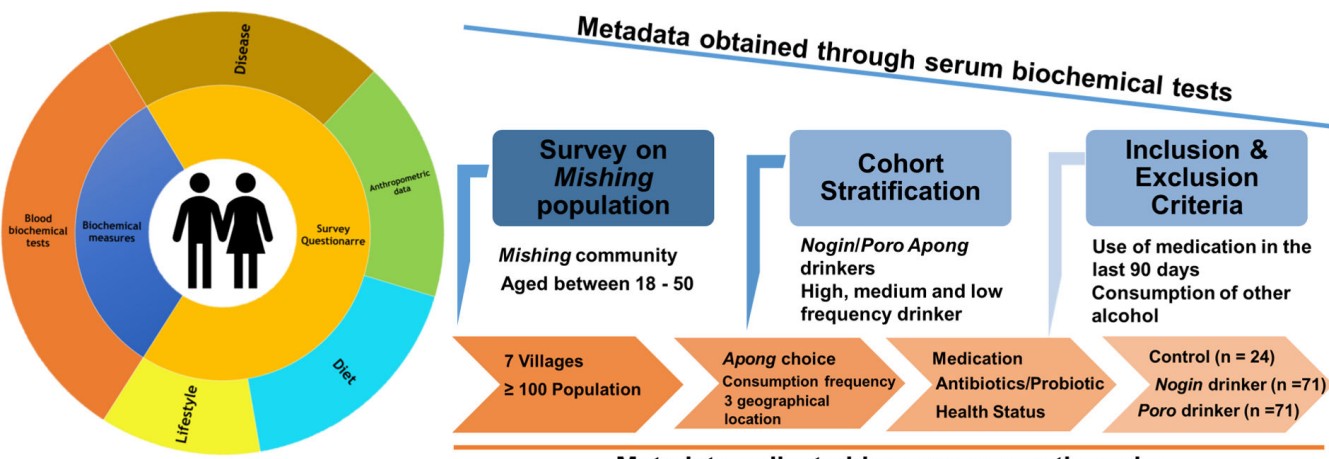

FIG 1 Summary of the cohort. The left half of the figure describes the metadata collected during the collection of samples. The right side describes the strategy used for sample collection; a survey was conducted on seven villages inhabited by the Mishing population to screen the prospective volunteers, later based on the inclusion, exclusion criteria, and other factors, and a total of 166 volunteers were retained for final analysis.

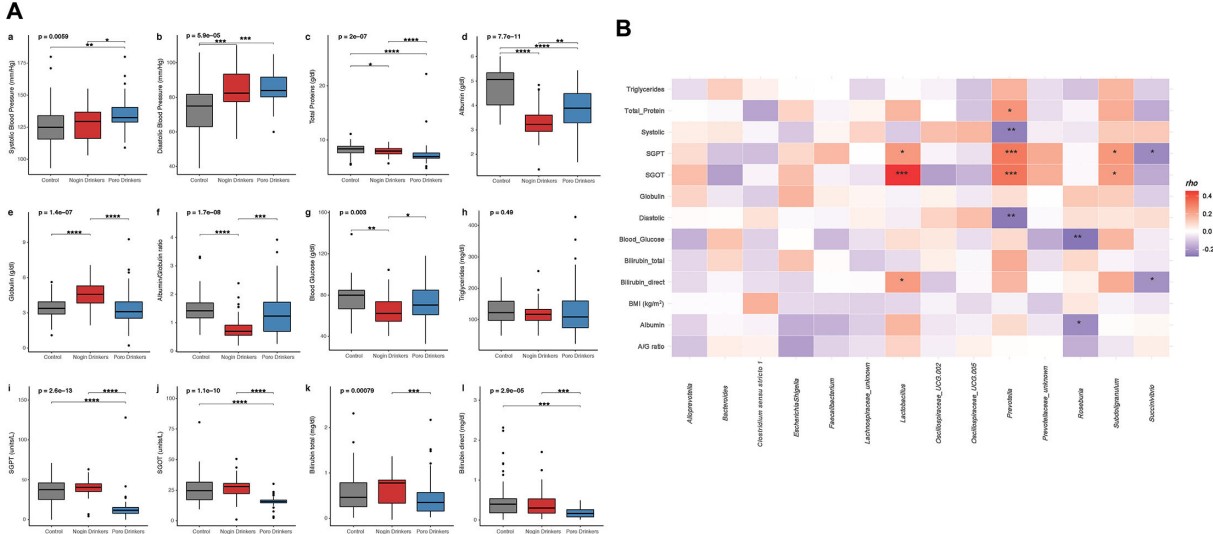

**FIG 2** (A) A comparison of anthropological factors and serum biochemical markers among Apong drinkers and non-drinkers: (a) systolic blood pressure, (b) diastolic blood pressure, (c) total protein, (d) albumin, (e) globulin, (f) albumin/globulin ratio, (g) blood glucose, (h) triglycerides, (i) SGPT, (j) SGOT, (k) bilirubin total, and (l) bilirubin direct. The probability of significance is denoted by *'s, where **** depicts a significance level of ≤0.0001, *** depicts ≤0.001, ** depicts ≤0.01, and * depicts ≤0.05. Only significant *P*-values are indicated. The *P*-values included here signify the global *P*-value as computed by the Kruskal-Wallis H test. (B) The association of anthropometric measures and serum biochemical with microbial signatures (genus level). The Spearman correlation values (rho) are depicted by a color scale from blue to red, where blue denotes a strong negative correlation and red denotes a strong positive correlation. The significant correlations are marked with *s, where *** depicts ≤0.001, ** depicts ≤0.01, and * depicts ≤0.05.

levels of bilirubin total and bilirubin direct (Fig. 2A, k and l). Although high bilirubin levels can indicate liver damage, low levels are not a health concern.

## The gut microbiota has a strong association with anthropometric markers and serum biochemical parameters

A strong association of the dominant taxa (at the genus level) was observed with the anthropometric markers and the serum biochemical markers. Among anthropometric markers, systolic and diastolic blood pressure were negatively correlated to *Prevotella* (rho = −0.26, *P* = 0.005 and rho = −0.27, *P* = 0.003, respectively). Liver markers, SGOT and SGPT, exhibited significant associations with the relative microbiome abundances at the genus level: SGOT exhibited positive correlations with *Prevotella* (rho = 0.32, *P* < 0.001) and *Lactobacillus* (rho = 0.24, *P* < 0.015). However, a negative correlation of SGOT was observed with *Butyrivibrio* (rho = −0.22, *P* = 0.029) and *Blautia* (rho = −0.32, *P* = 0.001). SGPT, on the other hand, correlated positively with *Prevotella* (rho = 0.32, *P* < 0.01) and *Lactobacillus* (rho = 0.45, *P* < 0.001), whereas negatively correlated with *Succinivibrio* (rho = −0.223, *P* = 0.02) (Fig. 2B). Other associations were observed with albumin (A), globulin (G), A/G ratio, bilirubin direct, and total protein (Fig. 2B).

## The gut microbiota composition varies between non-drinkers and Apong drinkers

We compared the gut microbial composition of Apong drinkers and non-drinkers. We found significant differences between Nogin and Poro drinkers and non-drinkers (Fig. 3A). Overall, the gut microbiota was dominated by *Bacteroidota* and *Bacillota* and by *Pseudomonadota* to a lesser extent. However, while *Bacteroidota* made up a significant proportion of the gut bacterial community in Nogin drinkers and non-drinkers, it made up a smaller proportion in Poro drinkers. The *Bacillota* to *Bacteroidota* ratio (erstwhile Firmicutes to Bacteroidetes F/B ratio), did not significantly vary among the Apong drinkers and the controls (Wilcox rank-sum test: *P* = 0.099 and 0.062, respectively) (Fig. S1). Notably, *Actinomycetota* was only detected in non-drinkers. Next, we identified taxa

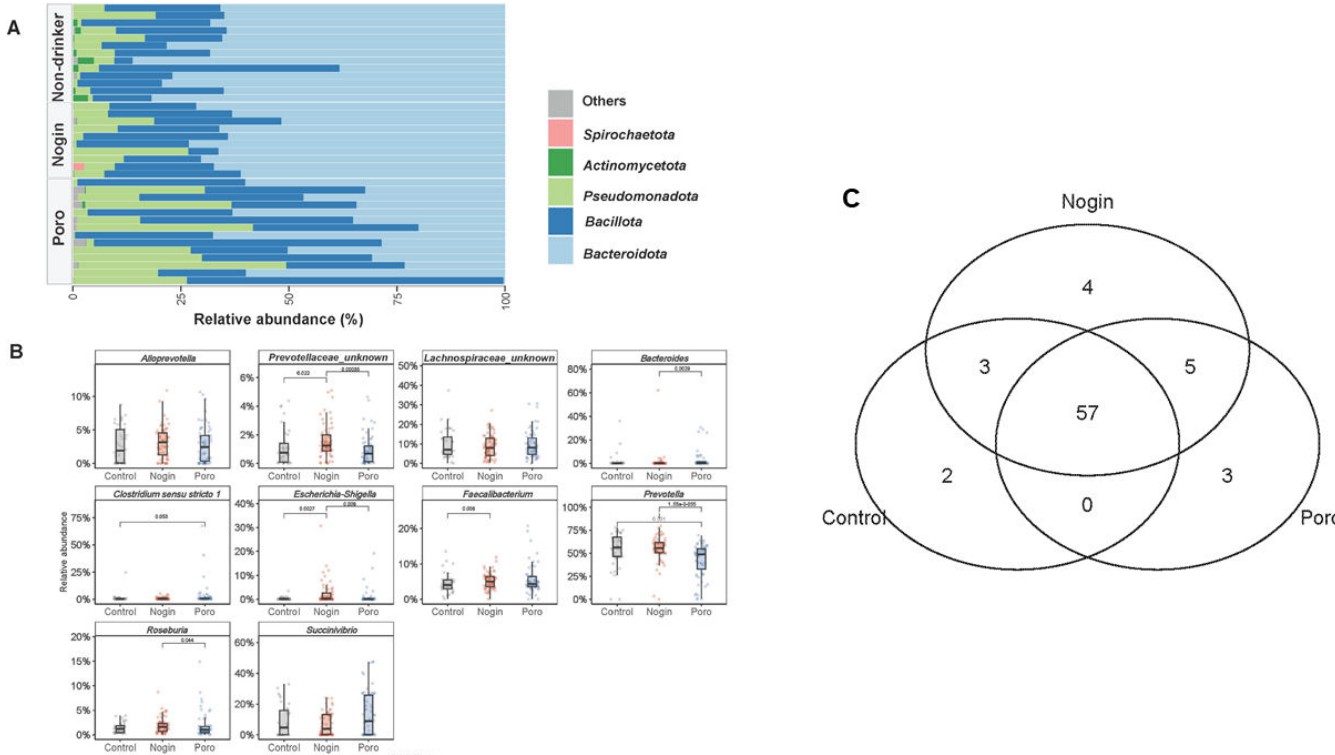

**FIG 3** (A) Relative abundance of microbial taxa at the phylum level in Apong drinkers and non-drinkers. (B) Top 10 microbial families that are differentially abundant among Apong drinkers and non-drinkers. Only significant *P*-values are indicated. (C) Venn diagram of microbial features (amplicon sequence variants) shared between different drink type groups or special to each group.

at the genus level that significantly differed in abundance among the Apong drinkers and non-drinkers (Fig. 3B). Apong drinkers had higher relative abundance of *Blautia*, *Dorea, Lachnospiraceae UCG-008,* and [*Ruminococcus*] *gnavus*, while *Bifidobacterium* was significantly higher in abundance among the non-drinkers. On the other hand, Poro drinkers had significantly higher abundance of *Butyricoccus, Christensenella, Coprobacter, Flavonifractor, Holdemania, Oscillospira,* and the genus UBA 1819 from the *Ruminoccaceae* family than Nogin drinkers and non-drinkers (Fig. S2). However, *Prevotella* was significantly lower among the Poro Apong consumers (Fig. 3B).

We also identified microbes that were specific to Apong drinkers (Fig. 3C). To do this, we identified and compared amplicon sequence variants (ASVs) that were detected in at least 50% of all participants at a minimum abundance of 0.1%. Most of the ASVs were shared (*n* = 57) among all participants and were part of the "core" microbiome. However, certain ASVs were only detected in Nogin (*n* = 4) or Poro drinkers (*n* = 3). Poro-specific ASVs were from the *Agathobacter*, *Oscillibacter,* and *Turicibacter* genera. The majority of the Nogin-specific ASVs were from the *Eubacterium ventriousun* group and *Howardella* genus, with the remaining two ASVs belonging to the members of the *Bacteroidales* order. The two ASVs that were detected in non-drinkers but not in Apong drinkers were from the *Bifidobacterium* and *Prevotella* genera.

## Gut microbial diversity in Apong drinkers is higher than in non-drinkers, and high-frequency Nogin drinkers have lower gut microbial diversity than other Nogin drinkers

We estimated the gut microbial diversity in Apong drinkers and non-drinkers using three different diversity metrics (Fig. 4A). Gut microbial diversity was significantly higher in the Apong drinkers than the non-drinkers.

We divided the participants into three categories based on their Apong consumption frequency: high (HD), medium (MD), and low (LD). Poro consumption frequency did not have a significant effect on microbial diversity (Fig. S3). However, among Nogin drinkers, high-frequency drinkers had significantly lower gut microbial diversity.

## Apong consumption and frequency have a significant effect on gut microbial composition

We investigated microbial composition between participants by computing Bray-Curtis and weighted UniFrac beta diversity (Fig. 4). The distance matrices revealed small but significant differences among the non-drinkers, Nogin drinkers, and Poro drinkers, as well as among subgroups of Apong drinkers based on the frequency of consumption (Fig. 4A).

Although location also had a significant effect on microbial composition, Apong usage had a larger overall effect (Fig. 4A). When blocking "location" in the permutational multivariate analysis of variance (PERMANOVA), drink type (non-drinker vs Nogin, or Poro drinker) still explained a significant portion of the variation in microbial composition (Bray-Curtis: $R^2 = 0.034$; $P = 0.003$ and Unifrac: $R^2 = 0.035$; $P = 0.001$). Despite inhabiting the same location (Majuli), the gut microbial composition of non-drinkers ($n = 24$) and Nogin drinkers ($n = 18$) formed two different clusters of principal coordinate analyses (PCoAs) ($R^2 = 4.79$; $P = 0.005$) (Fig. S4). Although many of the Apong drinkers ($n = 122$) consumed red meat occasionally, whereas the non-drinkers did not, red meat consumption and gender do not significantly explain variation in microbial composition (Bray-Curtis, $R2 = 0.019$; $P = 0.29$; weighted Unifrac, $R2 = 0.023$; $P = 0.19$) and (Bray-Curtis, $R2 = 0.00503$; $P = 0.77$; weighted Unifrac, $R2 = 0.00577$; $P = 0.3969$), respectively.

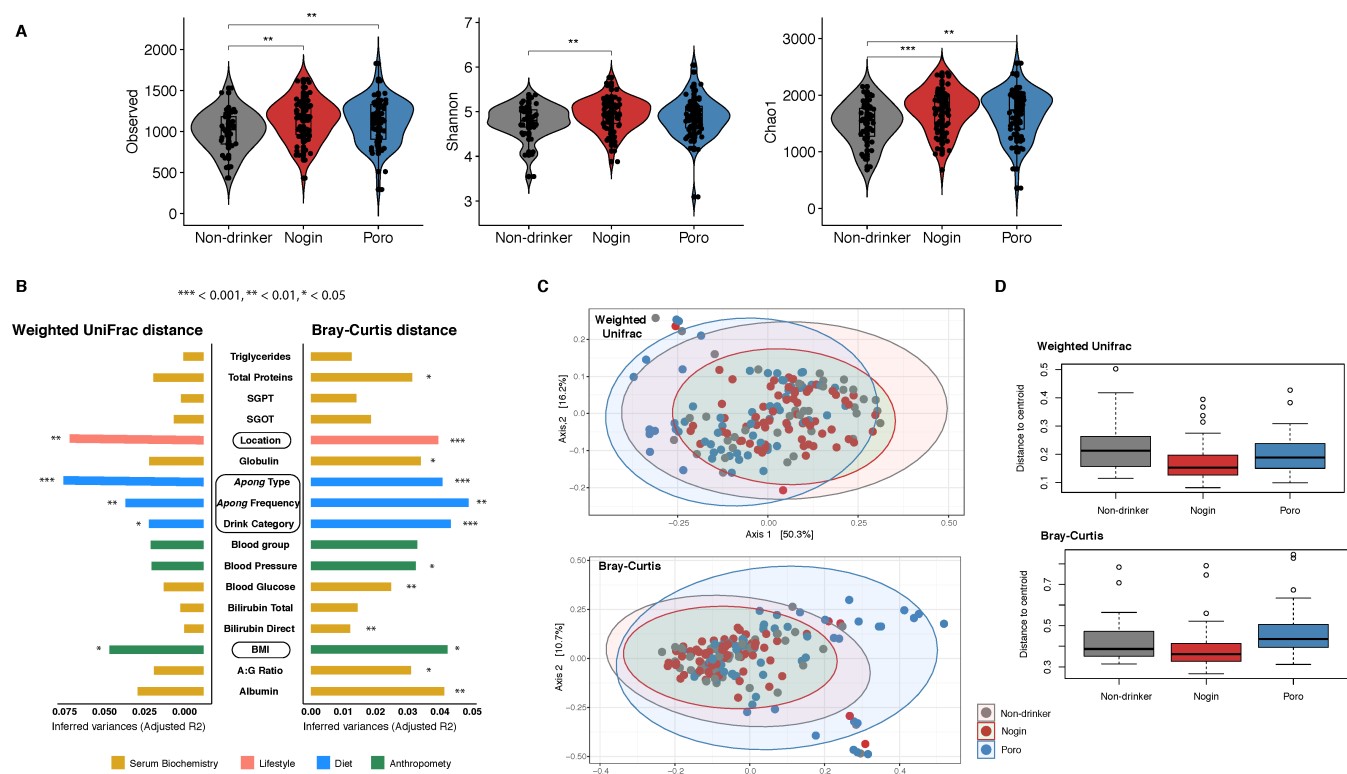

FIG 4 (A) Microbial diversity of Apong drinkers and non-drinkers as estimated by ASV richness and Shannon and Chao1 indices. (B) The inferred variance (adjusted $R^2$) is explained by each identified covariate as determined by permutational multivariate analysis of variance, calculated based on weighted UniFrac and Bray-Curtis dissimilarities. Statistically significant covariates with an adjusted $P < 0.05$ using the Benjamini–Hochberg (BH) method are shown. (C) Principal coordinate analysis (PCoA) of the weighted UniFrac and Bray-Curtis distances of the gut bacterial composition of Apong drinkers and non-drinkers. (D) Dispersion (i.e., distance to the centroid of the groups) of each drink type group in the PCoA plots.

Notably, blood serum biochemistry markers, such as total protein, albumin and globulin levels, and blood glucose and pressure, explained significant variation in the presence or absence of certain microbes, but not in the phylogenetic diversity of gut microbial composition.

## Nogin drinkers have a more homogeneous gut microbial community than Poro drinkers and non-drinkers

The gut microbial community of Nogin drinkers clustered together in the PCoA based on both weighted UniFrac and Bray-Curtis distances (Fig. 4C). However, non-drinkers and Poro drinkers had more heterogeneous microbial communities than Nogin drinkers. This was also demonstrated by the higher distance to the centroid (permutest in R, $n = 999$, $P = 0.03$) (multivariate homogeneity of groups dispersions) (Fig. 4D), where Nogin drinkers had the lowest distance to the centroid, so the highest homogeneity whereas Poro drinkers had a high distance, so more heterogeneity in the weighted UniFrac distance (considers phylogenetic relatedness).

## Fecal microbial metabolite profiles are different between the Apong drinkers and non-drinkers

To study the metabolic activity in the gut ecosystem of the cohort, an untargeted metabolite profiling with gas chromatography-mass spectroscopy (GC-MS) analysis was performed. Metabolites of microbial origins were identified using the human metabolome database (HMDB) for subsequent analysis. We extracted a total of 384 metabolites which comprised mainly amino acids, bile acids, fatty acids, indoles, and saccharides. Apong consumption led to the depletion of certain metabolites, such as acetamide, benzestrol, butanedioic acid, and cyclopropane carboxylic acid, while Poro drinkers had higher levels of undecanoic acid than non-drinkers and Nogin drinkers (Fig. 5).

We agglomerated gut microbes at the genus level and correlated these families to fecal metabolites using the mbOmic package in R (21). We found 216 significant correlations (adjusted $P$-values < 0.05) between microbial families and fecal metabolites. Only 28 of these correlations had a rho value higher than 0.70 (Table S2). The top 10 correlations are listed in Fig. 6A, and the majority of these correlations were with *Clostridium sensu stricto 1*, *Lachnospira,* and *Ruminococcus.*

Although having relatively heterogeneous gut microbial composition among participants (Fig. 4C and D), Poro drinkers had a highly uniform composition of fecal metabolites (Fig. 6B). Drink type (non-drinker, Nogin drinker, or Poro drinker) had an even larger effect on metabolite composition than on gut microbiota composition (Fig. 6B). Furthermore, we checked the association of anthropometric and serum biomarkers to the composition of the gut metabolome. Both systolic and diastolic blood pressure ($R^2 = 0.03029$ and $0.03238$ and $P = 0.027$ and $0.005$, respectively) explained a significant amount of variation in metabolite composition (PERMANOVA, the data were blocked for "location" of the samples).

Taken together, these results show that Apong drinkers had distinct blood serum markers, gut microbial composition, and fecal metabolites compared to non-drinkers.

## Apong consumers had lower levels of isovaleric acid among their fecal SCFAs compared to non-drinkers

The gut microbiota helps to break down undigested food through fermentation, producing short-chain fatty acids (SCFAs) as a result (22). The SCFAs are important for gut homeostasis and health and their levels are affected by diet. Although branched-chain fatty acids (BCFAs) may be important in the gut and could potentially serve as markers of gut microbial metabolism, they have received less attention than the major SCFAs (23).

We measured the levels of three major SCFAs and one BCFA (acetic acid, butyric acid, propionic acid, and isovaleric acid) in fecal samples from both Apong drinkers and non-drinkers using high-performance liquid chromatography (HPLC). Propionic acid was

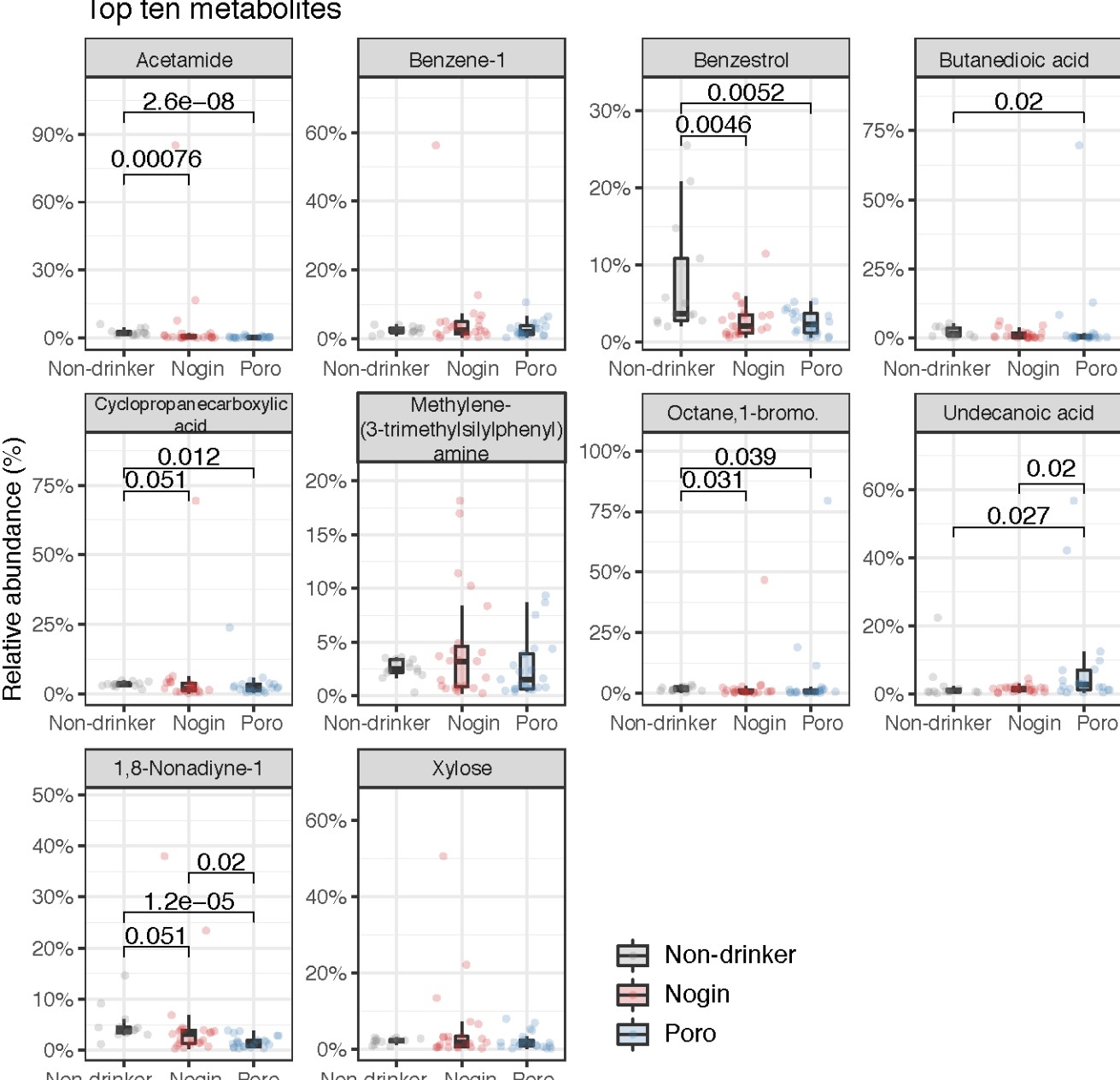

**FIG 5** Top 10 fecal metabolites that are differentially abundant among Apong drinkers and non-drinkers. Only significant *P*-values are indicated.

the most abundant SCFA in both groups, while butyric acid was the least (Figure 7). Some volunteers had very high levels of acetic acid, but it was not correlated with any other data (Fig. S5). Isovaleric acid, a BCFA considered to be harmful to the colon epithelium (24), was significantly lower in Apong drinkers compared to non-drinkers, but there was no significant difference for the other SCFAs (Fig. 7).

## DISCUSSION

The composition of the gut microbiome is influenced by various factors, including diet and lifestyle (4, 25–27), but the impact of a single component of diet within a population of the same ethnicity on the gut microbiome has not been studied before. This study investigates the effect of two types of traditional, rice-based fermented alcoholic beverages on the gut microbiome and health of the Mishing community in India.

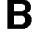

**A**

Top 10 correlating microbial taxa and metabolites

| taxa (family) | metabolite | rho | p value | adjusted p value |
|---|---|---|---|---|
| Clostridiaceae | Dimethylmalonic acid | 0.933214976 | 4.07E-26 | 1.40E-22 |
| Clostridiaceae | 1,6-Bis(2-propyn-1-yloxy)hexane | 0.932759101 | 4.88E-26 | 1.40E-22 |
| Selenomonadaceae | Decanoic acid | 0.92445005 | 1.08E-24 | 2.05E-21 |
| Clostridiaceae | Urea | 0.91609472 | 1.73E-23 | 2.47E-20 |
| Clostridiaceae | 5-(4H)-Oxazolones | 0.907582075 | 2.20E-22 | 2.51E-19 |
| Clostridiaceae | 2,4-Hexadien-1-ol | 0.881270757 | 1.51E-19 | 1.44E-16 |
| unknown Bacteroidota | Dodecanedioic acid | 0.876080983 | 4.58E-19 | 3.74E-16 |
| unknown Bacteroidota | Butanedioic acid | 0.864326188 | 4.72E-18 | 3.37E-15 |
| unknown Bacteroidota | Cyclopentane | 0.848294141 | 8.17E-17 | 5.19E-14 |
| unknown Bacteroidota | Trifluoroacetoxypentadecane | 0.832590054 | 9.87E-16 | 5.64E-13 |

**B**

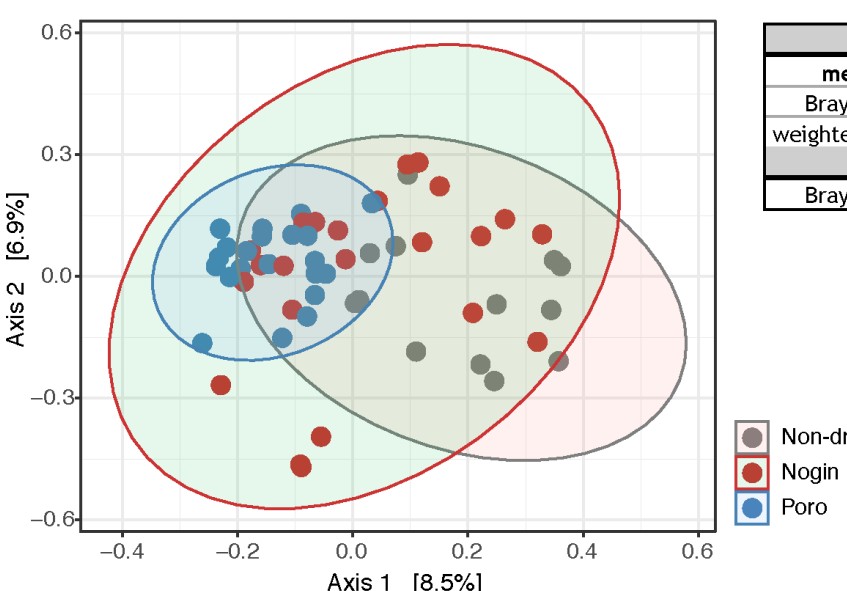

| microbial composition | | |
|---|---|---|
| metrics | explained variation | significance |
| Bray-Curtis | 3.45 | *** |
| weighted UniFrac | 6.259 | *** |
| metabolite composition | | |
| Bray-Curtis | 8.66 | *** |

FIG 6 (A) Top 10 highest correlations between gut microbial taxa at the genus level and fecal metabolites. (B) PCoA of the Bray-Curtis distances of the fecal metabolites of Apong drinkers and non-drinkers. The table explained variance by drink type (Nogo, Poro, and non-drinker) for microbial and fecal metabolites composition.

The monks were selected as the control group because there were no non-drinking individuals available from the same population. In addition, the proximity of the Satra to the sampling villages and their similar cultural practices made them the best group in the study area. However, there were some limitations for comparison with Apong consumers: the fact that only males were involved as monks in the Satra environment made it impossible to include female participants in the control group. However, Apong consumers included both males and females and the factor of gender did not explain a significant amount of variation in the composition of gut microbiota.

Despite the frequent consumption of alcoholic beverages, the volunteers in the study had normal body mass index (BMI) and healthy vital organ function. Previous research has shown that both varieties of Apong contain mild alcohol (9-11%) and have a high content of phenolics (18). Although the levels of total proteins, albumin, and blood glucose were within the normal range, individuals who consumed Apong had lower levels of these biomarkers compared to the non-drinkers. However, despite consuming

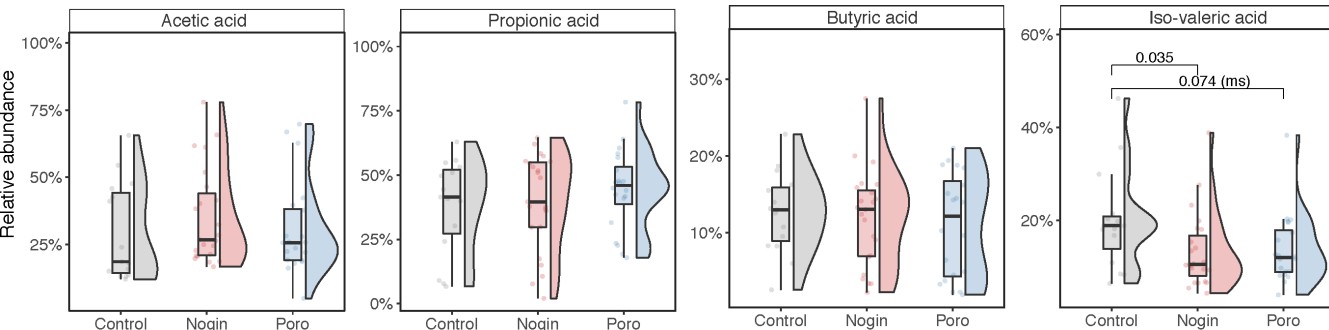

**FIG 7** Composition of the four SCFAs in the fecal samples of Apong drinkers and non-drinkers. Only significant *P*-values are indicated. "ms" denotes marginal significance.

Apong frequently and regularly, based on blood biomarkers, there was no evidence of any harmful effects on the liver.

Our study showed that the composition of the gut microbiome is affected by the choice of Apong. Apong consumers had a more diverse gut microbiome compared to the non-drinkers, which had a stable community with fewer ASVs. In addition, we also found that 57 ASVs were shared among the drinkers and non-drinkers, but 7 ASVs were unique to the Apong consumers alone (4 Nogin drinkers and 3 Poro drinkers). On examining the ASVs specific to the study groups, we observed that Nogin drinkers were enriched with *Eubacterium ventriousum* group and *Howardella*, while two of the unique genera belonged to the *Bacteroidales* order, whereas Poro drinkers were enriched with genera of *Agathobacter*, *Oscillibacter*, and *Turicibacter*.

The dominance of *Bacteroides* and *Lachnospira* in the gut microbiota has been linked to amino acids and complex carbohydrates in the diet (28). As Apong is rich in such components, it can be well speculated that consumption of Apong might have enriched such bacteria in the gut of consumers. The presence of *Prevotella* in all the study groups can be related to the abundance of rice (carbohydrates and green plants) in their diet (28, 29). Our findings suggest that a single dietary factor alone can significantly impact the modulation of the gut microbiome of a community, consistent with previous research on the gut microbiome of children in Burkina Faso (30). Our study also observed a strong association between variation in microbial composition and blood glucose levels and blood pressure, which is in accordance with previous findings (31–33). To our best knowledge, there is no study systematically analyzing the effect of alcoholic and/or sweet beverages on gut bacteria-related carboxylic acid levels in humans. However, it was reported that in the alcohol-treated mice, levels of carboxylic acids were less prominent than in the untreated group (34). However, further research is needed to understand the causal relationship between these factors and gut bacterial diversity, which may allow for the development of microbiome-based biomarkers for predicting lifestyle diseases. We have observed that individuals who consume Apong tend to have lower levels of certain carboxylic acids, including butanedioic acid (an isoform of succinate), although it does not have any modulatory effect on the ratio of *Bacillota* to *Bacteroidota* (erstwhile F/B ratio). Systemic circulation of gut microbiota-derived succinate has been linked to obesity and diabetes (35). However, consumption of Apong did not affect the levels of SCFAs, which are essential for maintaining colonic health and gut homeostasis. Previous studies have reported a decrease in fecal SCFAs following alcohol consumption, particularly with the consumption of rice-based alcoholic beverages (36–38). Furthermore, a human- and animal-based study from our laboratory highlighted that the consumption of commercial alcohols (whiskey) is associated with a decrease in *Prevotella* and fecal SCFAs and an increase in *Helicobacter* due to the non-ethanolic components (34). In the current study, no such observations were made. We speculate this might be due to the low alcoholic content of Apong and the presence of certain metabolites such as saccharides and amino acids in Apong. In addition, the

alcohol content of the two varieties of Apong (9–11% vol/vol) is similar; therefore, any effect due to the ethanolic component would be similar among the drinkers of both varieties.

In this study, the gut microbiome of the Mishing community was found to be dominated by *Prevotella*, a signature of the Indian population (39–43), which has been previously associated with a vegetarian or carbohydrate-rich diet (7). However, Poro drinkers had lower levels of *Prevotella* than non-drinkers and Nogin drinkers, although their gut microbiomes were colonized to a high extent by *Prevotella*. Furthermore, high blood pressure overall can be attributed to the dominance of *Prevotella,* which was found to be higher among Chinese hypertensive subjects (31).

Previous studies have reported that moderate alcohol consumption had a significant effect on cardiovascular health by affecting gut microbiota and serum metabolome (44). We found that the Apong drinkers had lower levels of *Actinomycetota* which was in line with previous reports on gut microbiota modulation, concerning consumption of moderate alcoholic beverages (44). Nevertheless, Apong consumption might play a pivotal role in the blood pressure regulation of its consumers, mediated through the microbiome. However, in-depth analyses are required to establish this fact.

Lastly, we found that the gut microbiome of the Mishing population was colonized to a high extent by *Succinivibrio*, a bacterium not previously reported in the Indian population (39–41). This bacterium is commonly found among hunter-gatherers and foragers (45, 46). The presence of *Succinivibrio* in the microbiome was previously reported in the Hadza hunter-gatherers and traditional Peruvian populations (45, 47). We speculate that the high abundance of *Succinivibrio* in the Mishing population may be due to co-habitation with domesticated animals. The co-presence of butyrate producers and other essential gut bacteria along with expected levels of SCFAs but lower levels of BCFA and blood serum measurements suggests that Apong does not have a detrimental effect on the structure and function of the gut microbiome. The approach of untargeted metabolite profiling using GC-MS provided qualitative information on the metabolites in terms of relative abundance rather than absolute abundance. Although informative, relative abundance data fail to provide a holistic scenario of the gut metabolome. However, the fecal SCFA and BCFA were quantified using standards that enabled us to quantify the fecal SCFAs.

## Conclusion

In conclusion, this study found that the choice of Apong, a traditional rice-based fermented alcoholic beverage, significantly modulates the gut microbiome composition and blood serum markers in the Mishing community in India. The gut microbiomes of Apong consumers were more diverse than those of non-drinkers, and Poro drinkers had lower levels of *Prevotella* than non-drinkers and Nogin drinkers. The gut microbiota of the Mishing community was also colonized by *Succinivibrio*, a bacterium not previously reported in the Indian population. The differences in gut metabolites between Apong drinkers and non-drinkers were even greater than those in the gut microbiome. These findings suggest that a single dietary factor can significantly impact the gut of a population and highlight the need for further research on the causal relationship between these factors and gut bacterial diversity for the development of microbiome-based biomarkers for predicting lifestyle diseases. This study being a cross-sectional study was unable to establish a causal relationship between Apong consumption and changes in the gut microbiome. However, we investigated the possible effects of Apong consumption on overall health by analyzing blood biomarkers and gut effects with data from different levels, including microbiome, metabolome, and SCFAs. We believe that our study provides valuable insights into the potential effects of Apong consumption on the gut microbiome and health of the Mishing community.

## MATERIALS AND METHODS

### Materials

All the reagents, media, and chemicals used in this study were of analytical grade. RNA-later solution (Cat. No RM49049), SCFAs standards viz. acetic acid (Cat. No. 5438080100), butyric acid (Cat. No 19215), isovaleric acid (Cat. No 78651), and propionic acid (Cat No. 94425) were procured from Sigma-Aldrich, USA. The QiAmp DNA stool mini kit (Cat. No: 19590) was procured from Qiagen, Inc., Germany. Blood collection vials, K3-EDTA vial (Cat. No 368860) and clot vials (Cat. No 368975), were procured from BD Diagnostics, Oxford, UK. Blood serum biomarker kits were procured from CCS Coral Clinical Systems, Tulip Diagnostics (P) Ltd., Goa, India.

### Study sites, volunteers, and sample collection

In this study, potential volunteers were identified through an electoral database and surveyed in locations dominated by the Mishing community. The control group consisted of individuals who do not consume Apong, 24 monks from a Vaishnavite satra, but follow a similar dietary pattern, while the experimental group included 142 individuals from the Mishing community with only slight differences in dietary habits. The volunteers recruited in our study were healthy, from the same ethnicity and region, and followed similar dietary habits except for occasional consumption of meat and regular consumption of Apong. This difference in diet between the Apong consumers and the control group was primarily due to religious practices. The volunteer's diet includes rice, green leafy vegetables, fish, and Apong (either Poro or Nogin based on their personal preferences) along with occasional consumption of meat and pulses. On the other hand, the control group which includes a group of monks from a Vaishnavite Satra does not consume alcohol and meat due to religious grounds, albeit other dietary patterns remain the same.

A bilingual survey questionnaire was used to collect information on dietary patterns, age, sex, medical history, family lineage, and demographics. Inclusion criteria were age (18–50 years) and ethnicity, while exclusion criteria were the consumption of antibiotics for the past 90 days, the use of health supplements and other drugs, consumption of any other liquor except Apong, and medical history. The Mishing population was stratified based on the amount of rice beer consumed per week, with non-drinkers classified as those who did not consume Apong, low drinkers as those who consumed less than 250 mL, medium drinkers as those who consumed 250–500 mL, and high drinkers as those who consumed more than 500 mL per day.

Fecal and blood samples were collected from the Mishing population living in the Telam, Dhemaji, and Majuli districts of Assam. A customized kit was used to collect fecal samples, with one container, including 2 mL of RNA later solution, used for DNA extraction and another for metabolomics studies. Three milliliters of blood was withdrawn from the volunteers by a phlebotomist. Blood was collected in a K3-EDTA vial and clot vials to separate plasma and serum, respectively. The first defecation of the day was only considered a valid sample. All the stool and blood were collected in the morning prior to consumption of any food. Owing to difficulties in the collection of fecal samples for metabolomics, only a subset of samples could be collected for metabolomic analysis (control = 14, Nogin drinkers = 22, Poro drinkers = 21; total ($n$) = 57). Fecal and blood samples, anthropometric data, and other information were collected under strict medical supervision. All the samples were immediately frozen after collection and were transported to the laboratory in frozen condition. Fecal and blood samples were stored at −80°C until processed. In addition to the samples and questionnaires, anthropometric data such as height, weight, BMI, and blood group were also collected from the volunteers. The demographic information of the volunteers is presented in Table S1.

## Analysis of biochemical parameters of plasma and serum

The plasma samples were analyzed using standard biochemical assay kits [CCS Coral Clinical Systems, Tulip Diagnostics (P) Ltd., Goa, India], following the manufacturer's instructions. Serum glucose, HDL cholesterol, albumin, globulin, total protein, and liver function tests, for example, SGOT, serum glutamic pyruvic transaminase, and alkaline phosphatase (ALP), direct and total bilirubin contents were determined.

## DNA extraction from faecal sample

DNA extraction was performed within 72 h of fecal sample collection. The QiAmp DNA stool mini kit (Cat. No: 19590 Qiagen, Inc., Germany) was used for the extraction of metagenomic DNA from the fecal samples following the manufacturer's instructions. Briefly, a 400 µL of fecal sample was mixed with 1,400 µL of stool lysis buffer (ASL buffer) (provided with the kit) and incubated at 90°C for 10 min. The supernatant was collected after brief centrifugation at 13,000 rpm for 2 min, to which Inhibitex tablet and ProteinaseK (supplied with the kit) were added. After a short incubation of 10 min at 70°C, the mixture was centrifuged at 13,000 rpm for 3 min. The supernatant was collected in a filter with a silica column. The column was washed twice with a wash buffer (provided with the kit) and the bound DNA was eluted with a preheated elution buffer (supplied with the kit). The amount of dsDNA was quantified using a dsDNA estimation kit with a Fluorometer (Quantiflour, Promega, Madison, USA).

## Library preparation, 16S rDNA amplicon sequencing, and analyses

The V3-V4 region of the 16S rDNA was amplified using the primer pairs 341F and 805R (48). The indexing and library preparation of the amplified DNA fragments was carried out using Nextera XT library preparation and indexing kits according to the Illumina MiSeq protocol (49). DNA fragments were multiplexed and subjected to 2 × 300 bp paired-end sequencing in an Illumina MiSeq machine with the sequencing service provider (Macrogen Inc., Seoul, Republic of Korea).

## Bioinformatics analyses of the amplicon data set

The paired-end reads generated from Illumina sequencing were processed using the LotuS2 pipeline (50). Reads having less than 170 bases in length were filtered out from the analysis. In LotuS2, the DADA2 algorithm (51) was used to cluster sequences into ASVs. Using the options for LULU (-lulu) (52) and UNCROSS2 (-xtalk), sequence clusters were curated and refined. ASVs were aligned with Lambda (53) to SILVA 138 (54), to obtain taxonomic assignments for ASVs using the LotuS2 LCA algorithm. Otherwise, default options in LotuS2 were used. The sequencing primers were removed by -forwardPrimer and -reversePrimer options in the LotuS2 pipeline. LotuS2 pipeline uses a strict filtering criterion and the pipeline was run using the default parameters. Out of 28,083,099 reads, 24,049,359 reads were classified as "high quality" and "medium quality", and 4,033,740 as low by the pipeline, which clustered into 24,187 ASVs. However, of these, 16,637 ASV were removed as chimeras which accounted for 10,805,289 reads stemming from a chemistry-related issue during sequencing. Although we had to filter a significant number of reads, we had a sufficient number of high-quality reads per sample, which varied from 251,08 to 118,968, where the median of the number of reads was 89,916.

The pre-processed data were further analyzed with phyloseq package (55) in R (version 3.6.1). Samples were rarefied to the number of reads of the sample having the smallest sample size (25108) using the rtk tool (56) for diversity analysis. For the estimation and calculation of diversity indices, the vegan (57) package was used.

## Metabolomics analyses of fecal samples

The following techniques were used for the analyses of fecal samples.

## Preparation of fecal samples for metabolomics

The fecal samples were freeze-dried for 24 h prior to metabolomics analyses. Briefly, an aliquot of fecal samples was frozen at −80°C for overnight prior to freeze-drying. The frozen samples were then freeze-dried in a −84°C freeze-drying instrument (Labconco, USA) operated under vacuum mode. Tubes were then tightly sealed to avoid moisture consumption and stored at −80°C until processed.

## GC-MS analysis

Untargeted fecal metabolite profiles were determined by GC-MS analysis. A 40-mg freeze-dried fecal sample was extracted with 1 mL of HPLC grade methanol (Merck, Mumbai, India) and kept in a shaker overnight at 1,200 rpm. The sample was centrifuged at 10,000 rpm for 10 min at 10°C. The extract was then dried at room temperature using a vacuum desiccator and re-suspended in a mixture of 40 μL of pyridine and 20 mg/mL of methoxyamine hydrochloride. After a brief vortex, the solution was incubated at 30°C for 90 min and then derivatized with 20 μL of N-methyl-N-trimethylsilyltrifluoroacetamide with 1% trimethylchlorosilane (Merck, USA) at 70°C for 30 min. The sample was then centrifuged at 3,000 rpm for 5 min and used for GC-MS analysis.

Samples were run in a Shimadzu GC 2010Plus-triple quadrupole (TP-8030) system fitted with an EB-5MS column (length: 30 m, thickness: 0.25 μm, ID: 0.25 mm). A 1 μL of the sample was injected in splitless mode at 300°C using helium as carrier gas at a 1 mL/min flow rate. The oven program was set at 70°C initially and ramped at 1°C/min for 5 min up to 75°C. Subsequently, it was increased at 10°C/min up to 150°C, and held for 5 min, followed by increasing at the same rate up to 300°C and held for 5 min. The mass spectrometer was operated at a continuous scan from 45 to 600 m/z in the electron ionization (EI) mode at 70 ev with 200°C as the source temperature. Peak identification was performed using the National Institute of Standards and Technology library, USA, by matching the mass spectra.

## Quantification of fecal SCFAs by RP-HPLC analysis

Fecal samples were lyophilized, and 100 mg of each sample was dissolved in a solvent prepared using acetonitrile (Cat. No 271004 Merck Millipore, Germany) and 10 mM $KH_2PO_4$ (pH 2.4) (Cat. No P5379, Sigma Aldrich, USA) in 1:1 ratio. The solutions were then vortexed vigorously for 5 min to ensure proper mixture and centrifuged at 4,000 rpm for 5 min. The supernatant was collected and filtered through a 0.22-μm syringe filter for downstream analysis.

Quantification of the SCFA and BCFA was carried out in an analytical HPLC instrument (Waters, USA) with 5 μm ODS2 (4.6 × 250 mm, Waters SPHERISORB) reversed-phase C18 analytical column. Two HPLC grade solvents (solvent A was 10 mM $KH_2PO_4$, pH 2.4 with phosphoric acid, while solvent B was 100% acetonitrile) were used in a gradient system with a flow rate of 1.5 mL/min. The absorbance of the eluted compound was monitored at 210 nm by the Photodiode array detector (PDA).

## Metabolomics data analyses

The GC-MS peaks were identified by matching the mass spectra with the National Institute of Standards and Technology (NIST), USA. Noisy peaks and column bleeds were removed from the metabolite list prior to downstream analysis. Functional annotation of the metabolites was performed with an online HMDB and Kyoto Encyclopedia of Genes and Genomes database. Further to quantify the fecal SCFAs and BCFA, standard solutions of acetic acid, butyric acid, propionic acid, and isovaleric acids were prepared within the concentration range of 100–800 ug/mL. These standard solutions were run in a Waters HPLC system in the same elution program to prepare the calibration curves. Organic acids were then quantified by extrapolating the value against the calibration curves.

## Statistical analyses

All the statistical tests were performed in the R platform using base functions and calling specialized packages such as phyloseq (55), vegan (57), microeco (58), microbiome (59), microbiomeutilities (60), and mbOmic (21). The normality distribution was determined by the Shapiro-Wilk test using the shapiro.test function of R. Comparisons among the anthropometric measures and serum biochemical markers were carried out using the Kruskal-Wallis test followed by pairwise Wilcoxon rank-sum test/Mann-Whitney U test with Bonferroni Hochberg correction to minimize the false discovery rate. To compare the microbial diversity among Apong drinkers and non-drinkers, the samples were first rarefied to an equal depth. Then, the alpha diversity indices were calculated for each sample, including the number of unique features (richness), the Shannon diversity, and the Chao1 metric. To determine whether there were significant differences in microbial diversity among Apong drinkers and non-drinkers, a Kruskal-Wallis test was performed. Before calculating the beta distance (Bray-Curtis and weighted UniFrac), the table was normalized to relative abundances. "adonis" and "mantel" functions in the vegan package were used to run PERMANOVA and Mantel tests to calculate metadata variables explaining variation based on beta diversity distances. "betadisper" function in the vegan package was used to estimate the homogeneity of drink types groups in the PCoAs. Multivariate analysis of the GC-MS-based metabolite data was performed within the MetaboAnalyst4 package (61). The metabolite data were normalized by the sum method to minimize the possible differences in concentration between samples followed by log transformation. Scaling of the data was performed using mean centering and division by the square root of the standard deviation of each variable to give all the variables equal weight regardless of their absolute value as available in the package. Relative abundances (%) were calculated by dividing the peak area of a compound by the total area of compounds identified in the same sample. The "corr" function from the mbOmic package was used to calculate Pearson correlations between the metabolomics data and microbial taxa at the genus level and $P$-values were adjusted using the Benjamini-Hochberg procedure for multiple comparisons.

## ACKNOWLEDGMENTS

The authors acknowledge the Institutional Level Biotech Hub (DBT, Govt. of India) and the Central instrumentation facility of IASST for providing the facilities. The authors also acknowledge Mr. Pinku Rajbongshi for assisting in the collection of blood samples. We are thankful to Dr. Dibyayan Deb, Dr. Atanu Adak, Mr. Prashanta Das, Mr. Dhrubajyoti Regon, Mr. James Doley, Mr. Deepak Mili, and Mr. Samujjal Saikia for assistance in the recruitment of volunteers and sample collection.

This research was funded by the Department of Biotechnology (DBT) under the Unit of Excellence Project (BT/550/NE/U-Excel/2014) and the SC/ST community development program in IASST (SEED/TITE/2019/103) funded by DST, Govt. of India. S.D. is thankful to IASST for providing financial support to carry out research work. E.O. and F.H. were supported by the European Research Council H2020 StG (erc-stg-948219, EPYC) and by the Biotechnology and Biological Sciences Research Council (BBSRC) Institute Strategic Programme Food Microbiome and Health BB/X011054/1 and its constituent project BBS/E/F/000PR13631, Core Capability Grant BB/CCG1720/1 and the work delivered via the Scientific Computing group, as well as support for the physical HPC infrastructure and data center delivered via the NBI Computing infrastructure for Science (CiS) group. F.H. was also supported by the European Union's Horizon 2020 Research and Innovation programme (H2020-EU.3.2.2.3., FNS-Cloud) grant Agreement No. 863059.

## AUTHOR AFFILIATIONS

[1]Life Sciences Division, Institute of Advanced Study in Science and Technology (IASST), Guwahati, Assam, India

²Department of Molecular Biology and Biotechnology, Cotton University, Guwahati, Assam, India
³Gut Microbes and Health, Quadram Institute Bioscience, Norwich, United Kingdom
⁴Digital Biology, Earlham Institute, Norwich, United Kingdom

## AUTHOR ORCIDs

Santanu Das  http://orcid.org/0000-0002-9798-6990
Ezgi Özkurt  http://orcid.org/0000-0002-6643-240X
Tulsi Kumari Joishy  http://orcid.org/0000-0003-3210-7622
Falk Hildebrand  http://orcid.org/0000-0002-0078-8948
Mojibur R. Khan  http://orcid.org/0000-0002-6909-1479

## FUNDING

| Funder | Grant(s) | Author(s) |
|---|---|---|
| Department of Biotechnology, Ministry of Science and Technology, India (DBT) | BT/550/NE/U-Excel/2014 | Mojibur R. Khan |
| Department of Science and Technology, Ministry of Science and Technology, India (DST) | SEED/TITE/2019/103 | Mojibur R. Khan |
| EC | European Research Council (ERC) | erc-stg-948219 EPYC | Falk Hildebrand |
| UKRI | Biotechnology and Biological Sciences Research Council (BBSRC) | BB/r012490/1, BBS/e/F/000Pr10355, BB/CCG1720/1 | Falk Hildebrand |

## AUTHOR CONTRIBUTIONS

Santanu Das, Conceptualization, Formal analysis, Investigation, Methodology, Visualization, Writing – original draft, Writing – review and editing | Ezgi Özkurt, Investigation, Methodology, Visualization | Tulsi Kumari Joishy, Methodology | Falk Hildebrand, Software, Writing – review and editing | Mojibur R. Khan, Conceptualization, Funding acquisition, Resources, Supervision, Writing – review and editing.

## DATA AVAILABILITY

Sequencing data are available on the NCBI SRA server under the BioProject ID PRJNA906264.

## ADDITIONAL FILES

The following material is available online.

### Supplemental Material

**Fig. S1 (mSystems00745-23-s0001.pdf).** *Bacillota/Bacteroidota* ratio.
**Fig. S2 (mSystems00745-23-s0002.png).** Differentially abundant bacteria.
**Fig. S3 (mSystems00745-23-s0003.pdf).** Alpha diversity between LD, MD, and HD.
**Fig. S4 (mSystems00745-23-s0004.pdf).** PCoA of the weighted UniFrac and Bray-Curtis distances of the gut bacterial composition of Apong drinkers and non-drinkers.
**Fig. S5 (mSystems00745-23-s0005.pdf).** Composition of the four short chain fatty acids (SCFAs).
**Supplemental text (mSystems00745-23-s0006.docx).** Legends for supplemental figures and tables.
**Table S1 (mSystems00745-23-s0007.xlsx).** Demographic information of the participants.
**Table S2 (mSystems00745-23-s0008.xlsx).** Correlations between gut microbial taxa at genus level and fecal metabolites.

## Open Peer Review

**PEER REVIEW HISTORY (review-history.pdf).** An accounting of the reviewer comments and feedback.

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
