## [Reviewer comments · mSystems]

A single dietary factor, daily consumption of a fermented beverage, can modulate the gut bacteria and faecal metabolites within the same ethnic community

Santanu Das, Ezgi Özkurt, Tulsi Joishy, Ashis Mukherjee, Falk Hildebrand, and Mojibur Rohman Khan

Corresponding Author(s): Mojibur Rohman Khan, Institute of Advanced Study in Science and Technology

Review Timeline:

Submission Date:	July 18, 2023
Editorial Decision:	August 22, 2023
Revision Received:	September 11, 2023
Accepted:	September 20, 2023

Editor: Paul Cotter

Reviewer(s): The reviewers have opted to remain anonymous.

Transaction Report:

DOI: <https://doi.org/10.1128/msystems.00745-23>

August 22, 2023

Dr. Mojibur Rohman R Khan
Institute of Advanced Study in Science and Technology
Division of Life Science
Vigyan Path, Paschim Boragaon
Garchuk
Guwahati, Assam 781035
India

Re: mSystems00745-23 (A single dietary factor, daily consumption of a fermented beverage, can modulate the gut bacteria and faecal metabolites within the same ethnic community)

Dear Dr. Mojibur Rohman R Khan:

Thank you for submitting your manuscript to mSystems. We have completed our review and I am pleased to inform you that, in principle, we expect to accept it for publication in mSystems. However, acceptance will not be final until you have adequately addressed the reviewer comments.

Preparing Revision Guidelines

Please return the manuscript within 60 days; if you cannot complete the modification within this time period, please contact me. If you do not wish to modify the manuscript and prefer to submit it to another journal, please notify me of your decision immediately so that the manuscript may be formally withdrawn from consideration by mSystems.

Sincerely,

Paul Cotter

Editor, mSystems

Journals Department
Reviewer comments:

Reviewer #1 (Comments for the Author):

The authors have done an excellent job of addressing all previously provided remarks on the manuscript, including to statistical analyses and descriptions.

Reviewer #3 (Comments for the Author):

The paper has made several improvements from the previous version and has provided much greater detail on the shaping of the gut microbiome with introduction of a fermented alcoholic beverage. however the following questions need further clarification

1. 50 % of the data being assigned as chimeric is too high to qualify for acceptance without clarification. Is this because the default parameters were used? were the primers removed and checked for their removal via cutadapt pipeline? at what stage does the quality drop (phead<30) for the average of all forward and reverse reads and then for the merged forward and reverse reads. This information should also be integrated into the manuscript.
2. if there are 216 correlations between gut microbiota and fecal metabolome then why is the supplementary only restricted top 30?
3. In the supplementary table 2 under the column labelled metabolites and under row number 30: metabolite "Holdemanella" is mentioned. please clarify and correct the table. Kindly double check the analysis given the nature of error in the table
4. lines 304 to 308: discussion of impact of apong in modulating blood pressure via the gut microbiome. how is this connection made. the cited literature talks about a multitude of long chain fatty acids that could be a potential impicator of CAD, however those metabolites are not seen or mentioned in this study...maybe further teasing apart can be done in the discussion section regarding the associations between microbial players and clinical conditions.

Minor comments:

1. being on a vegetarian diet might alter the blood albumin level and must be mentioned in the study
2. line 173:"We agglomerated gut microbes at the family level" however the results have been modified according to previous comments to genus level. this correction has to be made in the text appropriately
3. the discussion section mentions the correlation between apong and fecal carboxylic acid but again does not shed light as to what this could mean? have other studies reported this?

Reviewer #3

The paper has made several improvements from the previous version and has provided much greater detail on the shaping of the gut microbiome with introduction of a fermented alcoholic beverage. however the following questions need further clarification

Q1. 50 % of the data being assigned as chimeric is too high to qualify for acceptance without clarification. Is this because the default parameters were used? were the primers removed and checked for their removal via cutadapt pipeline? at what stage does the quality drop (phread<30) for the average of all forward and reverse reads and then for the merged forward and reverse reads. This information should also be integrated into the manuscript.

Response: *Thank you for the valuable suggestion. We removed the primers using the -forwardPrimer and -reversePrimer options in the LotuS2 pipeline (<https://doi.org/10.1186/s40168-022-01365-1>) and the pipeline reported in their detailed log file that primer removal was successful. LotuS2 pipeline uses a strict filtering criterion. 24,049,359 of reads were classified as “High-quality” and “Medium quality”, and 4,033,740 as low by the pipeline. We believe that relatively high proportion of chimeric reads (10,805,289 accounting for 44.43%) stemming from a chemistry-related problem occurred during sequencing. However, although we had to filter an important number of reads, our samples were deeply sequenced, therefore we still ended up with high number of reads: 24,049,359 reads (median: 89916 reads/sample). (Please see Line 415-418 for further information).*

The manuscript has been modified as below

Old manuscript	New manuscript	New Page number and Line number
Absent	The sequencing primers were removed by -forwardPrimer and -reversePrimer optionswhere the median of the number of reads was 89916	Page No 19 Line Nos. 466 to 472

2. if there are 216 correlations between gut microbiota and fecal metabolome then why is the supplementary only restricted top 30?

Response: *Thank you for the valuable comment. Although we obtained a total of 216 correlations, we had set a cut off limit at $\rho \geq 0.7$ to include only the strong correlations. Therefore, correlations having a strength of $\rho \geq 0.7$ were represented in the supplementary table 2. Accordingly, this has been changed in the text of the manuscript.*

The manuscript has been modified as below:

Old manuscript	New manuscript	New Page number and Line number
All the correlations are represented in Supplementary Table 2	Only 28 of these correlations had a rho value higher than 0.70 (Supplementary Table 2).	Page No 9 Line Number 175 and 176

3. In the supplementary table 2 under the column labelled metabolites and under row number 30: metabolite "Holdemanella" is mentioned. please clarify and correct the table. Kindly double check the analysis given the nature of error in the table

Response: *Thank you for the valuable comment. We apologize for the mistake in the table. The table has now been corrected.*

4.lines 304 to 308: discussion of impact of apong in modulating blood pressure via the gut microbiome. how is this connection made. the cited literature talks about a multitude of long chain fatty acids that could be a potential implicator of CAD, however those metabolites are not seen or mentioned in this study...maybe further teasing apart can be done in the discussion section regarding the associations between microbial players and clinical conditions.

Response: *Thank you for the valuable comment. We would like to clarify that the association between the gut microbes and the biomarkers (serum and anthropometric) were established through correlation studies. Moreover, in the discussion section we have drawn comparisons to studies those have similar findings. However, establishing a causal relationship of biomarkers and the gut microbes was beyond the scope of this study, which was mainly focused on understanding the role of a dietary factor. Nevertheless, as suggested we have expanded the discussion on how key gut bacterial members might play a role in the regulation of blood pressure.*

The manuscript has been modified as below

Old manuscript	New manuscript	New Page number and Line number
Absent	Elevated blood pressure has been found to be associated with the prevalence of bacterial genera like Prevotella , factors in influencing blood pressure regulation.	Page No. 12 and 13 Line Nos. 254 to 258
Absent	Furthermore, high blood pressure overall can be attributed towards the dominance of Prevotella , which was found to be higher among Chinese hypertensive subjects (31).	Page No. 14 Line Nos. 282 to 284

Minor comments:

Q1. being on a vegetarian diet might alter the blood albumin level and must be mentioned in the study

Response: *Thank you for your suggestions. Indeed dietary choices might influence the serum globulin level of an individual, especially for the vegetarians. Our primary objective in assessing biochemical parameters, such as serum globulin levels, was to gain insights into the functioning of vital organs like the liver and kidneys. We specifically aimed to determine if long-term consumption of Apong had any detrimental effects on these essential organs.*

Apong drinkers had similar dietary habits. Both Nogin and Poro drinkers consume occasionally meat. Although we observed significant difference in the serum globulin levels among the groups, it was within the permissible limit. We are pleased to report that our study did not reveal any adverse impacts of Apong on these vital organs.

Q2. line 173:"We agglomerated gut microbes at the family level" however the results have been modified according to previous comments to genus level. this correction has to be made in the text appropriately

Response: *We apologize for this. Following your suggestions now we have corrected the family level to genus level at appropriate place.*

The manuscript has been modified as below

Old manuscript	New manuscript	New Page number and Line number
We agglomerated gut microbes at the family level and correlated these families to fecal metabolites using the mbOmic package in R(21).	We agglomerated gut microbes at the genus level and correlated these families to fecal metabolites using the mbOmic package in R(21).	Page No. 9 Line Nos. 173 and 174

Q3. the discussion section mentions the correlation between apong and fecal carboxylic acid but again does not shed light as to what this could mean? have other studies reported this?

Response: Thank you for highlighting the importance of linking Apong consumption and faecal carboxylic acids. Following your suggestion, now we have explained this in the discussion section.

The manuscript has been modified as below:

Old manuscript	New manuscript	New Page number and Line number
Absent	To our best knowledge,	Page No 1

	there is no levels of carboxylic acids were less prominent than the untreated group.	Line Nos. 299 to 303
--	--	----------------------

September 20, 2023

Dr. Mojibur Rohman R Khan
Institute of Advanced Study in Science and Technology
Division of Life Science
Vigyan Path, Paschim Boragaon
Garchuk
Guwahati, Assam 781035
India

Re: mSystems00745-23R1 (A single dietary factor, daily consumption of a fermented beverage, can modulate the gut bacteria and faecal metabolites within the same ethnic community)

Dear Dr. Mojibur Rohman R Khan:

Your manuscript has been accepted, and I am forwarding it to the ASM Journals Department for publication. For your reference, ASM Journals' address is given below. Before it can be scheduled for publication, your manuscript will be checked by the mSystems production staff to make sure that all elements meet the technical requirements for publication. They will contact you if anything needs to be revised before copyediting and production can begin. Otherwise, you will be notified when your proofs are ready to be viewed.

If you would like to submit a potential Featured Image, please email a file and a short legend to msystems@asmusa.org. Please note that we can only consider images that (i) the authors created or own and (ii) have not been previously published. By submitting, you agree that the image can be used under the same terms as the published article. File requirements: square dimensions (4" x 4"), 300 dpi resolution, RGB colorspace, TIF file format.

We recognize that the video files can become quite large, and so to avoid quality loss ASM suggests sending the video file via <https://www.wetransfer.com/>. When you have a final version of the video and the still ready to share, please send it to mSystems staff at msystems@asmusa.org.

Sincerely,

Paul Cotter
Editor, mSystems

Journals Department
E-mail: mSystems@asmusa.org